# The Effect of Physical and Mental Health and Health Behavior on the Self-Rated Health of Pregnant Women

**DOI:** 10.3390/healthcare9091117

**Published:** 2021-08-28

**Authors:** Eunwon Lee, Jiyoung Song

**Affiliations:** 1Department of Nursing, Gwangju University, Jinwol-dong, Gwangju-si 61743, Korea; ewlee@gwangju.ac.kr; 2College of Nursing, Korea University, Anam-dong, Seongbuk-Gu, Seoul 02841, Korea

**Keywords:** pregnant women, self-rated health, health behaviors

## Abstract

Self-rated health (SRH) is an evaluation tool to assess an individual’s level of health, including both general health and personal experience. There have been existing studies on women’s SRH; however, few studies have been on pregnant women’s SRH and its associated factors. This study investigated the SRH of pregnant women and their factors using the Korea Community Health Survey. The chi-squared test and multivariable logistic regression were used to investigate the effects of demographic, physical, and mental health and health behaviors on the SRH of pregnant women. As a result of the study, 54.0% out of 1032 pregnant women had good SRH. Their SRH score was good when they were young, economically active, and living in cities. Poor SRH was observed with depression, hypertension, and after experiencing a fall. Good SRH was found when they exercised and slept for more than 8 h a day. This study is the first to observe the SRH of pregnant women and its related factors in South Korea.

## 1. Introduction

Childbirth can cause long-lasting health problems for women as they need to adapt to physiological processes and a new environment as a mother [1]. In addition, the importance of women’s health is emphasized in that the risk factors exposed to pregnant women are directly related to the health of the fetus and that the health of the mother affects the newborn and the rest of the family [2,3].

Pregnancy and childbirth are important processes within women’s health, with a large proportion of women experiencing this at least once in their lives. Therefore, maternal health during this period can be an opportunity to contribute to the promotion of women’s health [4,5]. The average birth age in Korea is rising faster than in European countries, and considering the high level of education of women, participation in economic activities, and advances in medical technology such as infertility treatment, the birth age is expected to increase [6]. As the childbearing age has recently increased, the interval between giving birth and entering middle age is now shorter [3]. Therefore, the health problems of pregnant and childbearing women should be considered significant. [6]. Hence, it is necessary to conduct a comprehensive evaluation of the overall health status of pregnant women by using self-rated health (SRH), which is a strong predictor of future disease morbidity and mortality [7,8]. 

SRH is one’s health condition recognized by an individual and is a measure used to predict health level and includes both objective health and personal experience [9,10]. SRH is a powerful indicator of general physical health, which allows the prediction of lifestyles such as physical activity, medical service use, prevalence, mortality, and well-being [8,11]. Therefore, it is widely used as a more effective and reliable indicator for predicting mortality than doctor’s assessments [12]. Self-rated health was useful in several studies. Mothers with poor SRH had many complications, such as anemia during pregnancy and postpartum bleeding [8]. In a study that confirmed SRH and physiological indicators of 101 women in the second trimester, it was reported that women with low SRH levels had higher serum interleukin-1β, an objective health indicator [9]. Considering these points, it will also be necessary to assess self-rated health from women in the pregnancy stage.

Factors influencing SRH, such as demographic and sociological characteristics, physical and mental health, and health behaviors, have been reported. According to previous studies, SRH levels were better in men than women [13,14,15] and in younger populations [9,16]. Women who are engaged in economic activities [13,17] and living in cities rather than rural areas [10,13] demonstrate a good level of SRH. Health behaviors, such as exercise and smoking [9,18], are also known factors affecting SRH. SRH has been reported to be poor even if there is a physical disease such as diabetes [2,19], depression [9,20], and a fall experience [18].

Studies on women’s SRH were mainly conducted on gender comparison [14,21] or international-immigrant married women [22,23]. Studies on the SRH of women of childbearing age [3,5,24] have been conducted, but not on pregnant women. As the average age of childbirth is rising in Korea, it is expected that women’s health problems might diversify and worsen; therefore, studies on the SRH of pregnant women are necessary. This study aims to prepare basic data for effective interventions for women’s health by investigating the effects of demographic factors, physical and mental health, and health behaviors on the SRH of pregnant women.

## 2. Materials and Methods

### 2.1. Study Participants

This study was based on the 2019 Korea Community Health Survey (KCHS) [25]. It was conducted within 255 health centers nationwide since 2008 for adults aged 19 or older to produce comparable regional health statistics to formulate and evaluate regional healthcare plans. In order to collect data, a trained investigator visited a household selected as a sample, provided explanations of the investigation and confidentiality to the person surveyed, received consent to participate, and conducted a 1:1 interview method survey. 

In this study, a pregnant woman was defined as a woman who answered ‘Yes’ to the question “Are you pregnant?”. Among a total of 222,099 participants, any exclusion criteria such as data from men, women over 50 years, or missing data regarding mental health and health behaviors were excluded from the analysis. The final participants totaled 1032 pregnant women, and they were categorized into two groups, the “good group” (*n* = 557) or “poor group” (*n* = 475), depending on the level of SRH. 

### 2.2. Study Variables

#### 2.2.1. Demographic Characteristics

The demographic characteristics used data on age (20–29, 30–39, 40–49), level of education, economic activities, and area of residence. The demographic characteristics were modified to suit the purpose of this study, based on the literature review [15,17,25]. Education levels were divided into three categories: ≤high school, college (2–3 years college), or ≥university (4 years college). Economic activities were defined by the question “In the past week, have you worked for 1 h or more for income purposes, or worked for 18 h or more as an unpaid family worker?” The question was divided into ‘yes’ or ‘no’. In the case of a residential area -dong, it was divided into city, and in case of -eup or -myeon, it was divided into rural.

#### 2.2.2. Self-Rated Health

SRH was assessed using a single item. Participants were asked to value their own health (How would you rate your health in general?) choosing from five possible answers: very good, good, fair, poor, or very poor. In this study, SRH was divided into “good” (very good or good) or “poor” (fair, poor, very poor) based on previous studies [8,22,26].

#### 2.2.3. Physical and Mental Health

Physical and mental health used mood of depression, falling experiences, hypertension, and diabetes. The mood of depression was evaluated as ‘yes’ or ‘no’ to the question, “Did you feel sad or desperate enough to interfere with your daily life for more than two weeks in a row in the last year?” The fall experience was divided into ‘yes’ and ‘no’ to the question “Have you ever fallen, including slipping, stepping, sinking, and falling in the last year?”. Finally, hypertension and diabetes were diagnosed by doctors, as per criteria.

#### 2.2.4. Health Behavior 

Health behavior characteristics used exercising, smoking, drinking, and sleeping. Based on the literature review [9,15,25], the health behavior characteristics were modified to suit the purpose of this study. In the exercise category, in the past week, on how many days did you perform vigorous physical activity such as running, fast swimming, or jumping rope for more than 10 min? If the question was answered at least once a day, yes, otherwise, no. In the smoking section, ‘non-smokers’ were classified as those who had never smoked cigarettes in their lives, ‘quit smoker’ was classified those who had smoked in the past but currently do not smoke, ‘smoker’ was classified as those who smoked every day and smoked sometimes. In the drinking section, a non-drinking group was classified as those who did not drink alcohol in the past year and the others were classified into ‘drinking group’. ‘Sleep’ was classified based on the number of hours slept (0–7 h vs. more than 8 h). 

### 2.3. Data Analysis 

Since the Korea Community Health Survey samples were drawn under a complex stratified sampling design, colonies and weights were generated according to the data analysis guidelines of the Korea Disease Control and Prevention Agency and then a composite sample analysis was used. According to the SRH, the characteristics of subjects were tested with frequency, percentage, and chi-square tests. In order to determine the effects of demographics, health behaviors, and physical health on SRH in pregnant women, multiple logistic regression analyses (OR) have been used to calculate the confidence interval (CI). For data analysis, the IBM SPSS statistics 23.0 program was used. All statistical significance levels were set to *p* < 0.05.

### 2.4. Ethical Considerations 

The Korea Community Health Survey is a government-designated statistical tool based on Article 17 of the Statistics Act (Approved No. 117015). The Korea Disease Control and Prevention Agency (KDCA) provided anonymous data so that individuals could not be identified from the survey data. The raw data used in this study were requested from the KDCA (https://chs.cdc.go.kr/chs/index.do, accessed on 25 May 2021) and received through the approval process for use. The research was conducted after receiving an exemption from the Public Institutional Review Board of the Ministry of Health and Welfare (IRB No.: P01-202105-22-008). 

## 3. Results

### 3.1. Demographic Characteristics

The demographic characteristics of a total of 1032 pregnant women in this study are shown in Table 1. Ages were 20 to 29 years old 25.5% (*n* = 263), 30 to 39 years old 61.0% (*n* = 630), and 40 to 49 years old 13.5% (*n* = 139). The average age was 33.4 ± 5.86. The level of education was 21.5% under high school graduation, 26.8% at college, and 51.7% after graduation from a university. 45.8% of the subjects were not engaged in economic activities, and 54.2% were engaged in economic activities. As for the residential area, 31.4% were rural and 68.6% were urban. The self-rated health level was 9.5% (*n* = 86) very good, 46.2% (*n* = 471) good, 39.7% (*n* = 434) fair, 4.3% (*n* = 39) poor, 0.4% (*n* = 2) very poor.

### 3.2. Characteristics of the Subject according to Self-Rated Health

As a result of the SRH survey of pregnant women, 54.0% (*n* = 557) of women answered good and 46.0% (*n* = 475) were found to be poor (Table 2). In terms of demographic characteristics, physical and mental health, and health behaviors, there were significant differences between the good and poor groups in the SRH, except for diabetes and drinking. 

Among pregnant women between the ages of 20 and 29, 55.1% of women answered that they had good SRH, 56.2% were between 30 and 39 years old, and 41.7% were between 40 and 49 years old. At the level of education, 50.0% of pregnant women who had graduated from high school or below had good SRH, 52.4% had graduated from two-year and three-year colleges, and 59.0% had graduated from a four-year college or higher. Among pregnant women who do not engage in economic activity, 60.9% of women responded that their SRH was good, and 51.7% of pregnant women were economically active. Among pregnant women living in rural areas, 49.3% of women answered that they had good SRH, and 56.9% of pregnant women living in cities said they had good SRH.

Among the pregnant women who felt depressed, 40.6% had good SRH and 56.7% were pregnant women who did not feel depressed. 39.2% of the pregnant women who had experienced a fall had good SRH, and 57.8% of the pregnant women had no experience of falling. In pregnant women with hypertension, 6.9% had good SRH, and 56.5% in pregnant women without hypertension. In pregnant women with diabetes, 42.2% of the subjects had good SRH, and 55.9% of those without diabetes were not significant (*p* = 0.12). 

Among pregnant women who exercised, 66.6% had good SRH and 54.8% were pregnant women who did not exercise. Among the smoking pregnant women, 25.2% of the group had good SRH, 46.6% of the smoking cessation group, and 56.6% of the non-smoker group. In the drinking group, 55.1% of the subjects had good SRH, and 56.0% of the non-drinking pregnant women were surveyed (*p* = 0.59). Among the pregnant women who slept for 0~7 h, 51.9% of the pregnant women had good SRH, and 59.2% of the pregnant women who slept for ≥8 h had good SRH.

### 3.3. Factors Affecting Self-Rated Health of Pregnant Women

Table 3 shows the effects of demographic characteristics, physical and mental health, and health behaviors on SRH.

The SRH of women aged 30 to 39 was 0.86-fold (95% CI: 0.73–1.00) and 0.46-fold (95% CI: 0.35–0.60) for women aged 40 to 49 compared to pregnant women aged 20 to 29. The SRH of pregnant women graduating from colleges was 0.91-fold (95% CI: 0.73–1.15) and 1.13-fold (95% CI: 0.95–1.40) above university graduation compared to those with a high school graduation or less. The SRH of pregnant women who engaged in economic activities was 1.59-fold (95% CI: 1.37–1.85) compared to those who did not engage in economic activities. Compared to pregnant women living in rural areas, pregnant women living in urban areas were 1.35-fold (95% CI: 1.17–1.56).

In pregnant women with depression, the SRH was 0.59-fold (95% CI: 0.44–0.78) compared to those without depression. In pregnant women who had experienced a fall, the SRH was 0.46-fold (95% CI: 0.37–0.56) compared to those who had not fallen. SRH in pregnant women with hypertension was 0.09-fold (95% CI: 0.02–0.42) than women without hypertension. The SRH of pregnant women diagnosed with diabetes was 1.08-fold (95% CI: 0.61–1.89) than that of undiagnosed women. Among pregnant women who exercised, the SRH was 1.92-fold (95% CI: 1.53–2.42) compared to those who did not exercise. The SRH of pregnant women in the smoking cessation group was 0.74-fold (95% CI: 0.54–0.97) and 0.55-fold (95% CI: 0.19–1.61) in the smoking group compared to non-smoking pregnant women. The SRH of women in the drinking group compared to non-drinking pregnant women was 0.93-fold (95% CI: 0.81–1.08). The SRH of pregnant women who slept for 8 h or more was 1.25-fold (95% CI: 1.10–1.43) compared to those who slept for 0 to 7 h.

## 4. Discussion

This study used the Korea Community Health Survey to investigate the SRH of pregnant women and identify influencing factors.

According to the study, 54.0% of 1032 pregnant women said they had good SRH. This result was compared to previous studies conducted with pregnant women in America, where good SRH was 39.6% [9], and 65.3% among women with experience of pregnancy in Congo [26]. These different results are thought to be due to the research being conducted in different environments. In addition, some studies have shown that the first birth age [1] or the number of births [1,26] does not affect the SRH of pregnant women. Considering this, further research is required in the future, not only in the whole population but also in various groups.

The subjects of this study were surveyed with an average age of 33.4 years, and the results of a previous study conducted on pregnant women in Korea [4] showed similar results to those of 33.2 years of age. In this study, pregnant women over 40 were also surveyed as 13.5% of the total subjects. Just as it is necessary to provide nursing care tailored to the health care needs of older pregnant women [6], individual personalized health care will also be required.

Previous studies have shown that old age [9,16,17] or low education levels [7,15] were related to bad SRH. However, in other studies, age or education level did not affect SRH [5,24]. In our study, neither age nor education levels affected SRH. In previous studies [13,16], the odds ratio of having good SRH for pregnant women who are engaged in economic activities was statistically higher than that of those who were not engaged in economic activities (95% CI: 1.37–1.85). In previous studies [5,26], meaningful effects were derived for employment type (part-time vs. full time) and each type of industry. Future studies will require research considering various working conditions, including employment type, type of business, working hours, and shift work. Pregnant women living in the city had a higher rate of good SRH than those living in the countryside [10,13]. In this study, the odds ratio of SRH in pregnant women living in cities was statistically higher (OR, 1.35, 95% CI: 1.17–1.56) than that of pregnant women living in rural areas. In the city, pregnant women can easily access medical institutions, as well as various cultural facilities, which is likely to have influenced this result. In this regard, policy considerations, such as the designation of vulnerable delivery areas, are necessary so that all pregnant women can experience medical care equality.

As shown in previous studies [9,20], pregnant women had poor SRH when they felt depressed. Counseling and intervention programs are necessary for promoting the health of pregnant women and for the health of fetuses. Good SRH is also reported if social support and social networks are good [27], and it will be important to use various resources in the community to decrease the rates of depression among pregnant women. However, this study has a limitation in that it used a single item to examine depression in pregnant women. In future studies, it will be necessary to examine the relationship between depression and SRH using a depression measurement tool with validity. 

SRH was poor if the women had experienced a fall [18]; this study showed a similar result. Maintaining a safe environment to prevent falls for the health of pregnant women is important. Pregnant women are advised to wear low-heeled shoes and avoid risk factors that may cause falls. SRH is said to be poor if there are comorbid conditions (e.g., hypertension, hyperlipidemia, diabetes mellitus, asthma, heart disease, and tuberculosis) [15]. This study also found that the SRH of pregnant women with hypertension is poor. Studies on the self-rated health effects of diabetes in pregnant women have not been clearly identified due to problems such as obesity, which is related to diabetes [2,19]. In this study, diabetes was also found to have no effect on the SRH of pregnant women. This study did not include many pregnant women that had been diagnosed with diabetes. Follow-up studies including many pregnant women with diabetes will be needed.

Exercise is known to be good for SRH [18]. This study also found that pregnant women who exercise have good SRH. Individual exercise suitable for pregnant women will need to consider the period of fetal pregnancy or fetal health conditions. Another study showed that smoking was not related to SRH [28]. However, the SRH was poor when comparing quit-smokers with non-smokers, like the previous study [9]. In addition, compared to the previous study, which showed a 2.4% smoking rate in pregnant women [29], this study found that only 1% (*n* = 13) smoked. Smoking cessation is challenging to achieve in a short period of time, so a variety of smoking education programs will be needed before adulthood. There is a limit to generalizing the results due to the small number of smoking groups in this study, and repeated studies will be needed in the future. As previous studies have shown that drinking does not affect SRH [15], drinking in this study did not affect the SRH of pregnant women. A prior study has shown that time slept does not affect SRH [16], but this study found that pregnant women’s SRH was good if they slept for more than 8 h compared to those who slept less than 7 h. In this regard, more studies that consider the total sleeping time of pregnant women are needed.

This study may be meaningful as it represents the SRH of pregnant women and its’ related factors in South Korea. However, there are some limitations. This study is a cross-sectional study using secondary data, and it is difficult to explain the temporal context of the SRH of pregnant women and their related factors. In addition, this study was conducted without considering the obstetric history, pregnancy history, and past history of pregnant women in detail. Therefore, we recommend that future studies consider the obstetrical information of pregnant women.

## 5. Conclusions

This study investigated the SRH of pregnant women and their influencing factors by using the Korea community health survey. According to our study, 54.0% of 1032 pregnant women had good SRH. The younger the pregnant women were, the better their SRH was if they were engaged in economic activities and lived in cities. Pregnant women with depression, fall experiences, and hypertension had poor SRH. SRH was found to be good when exercising and sleeping for more than 8 h. The results of this study will be used as basic data when developing education programs related to pregnant women’s health in the future. 

## Figures and Tables

**Table 1 healthcare-09-01117-t001:** Demographic characteristics of pregnant women (*n* = 1032).

Variables	Categories	*n* (%) or M ± SD *
Age	20–29	263 (25.5)
	30–39	630 (61.0)
	40–49	139 (13.5)
	total	33.4 ± 5.86
Education	≤High school	222 (21.5)
	College	277 (26.8)
	≥University	533 (51.7)
Economic Activity	No	473 (45.8)
	Yes	559 (54.2)
Residence	Rural	324 (31.4)
	Urban	708 (68.6)
Self-rated Health	Very Good	86 (9.5)
	Good	471 (46.2)
	Fair	434 (39.7)
	Poor	39 (4.3)
	Very Poor	2 (0.4)

* M ± SD, mean ± standard deviation.

**Table 2 healthcare-09-01117-t002:** Characteristics of the subject according to SRH (*n* = 1032).

Variables	Categories	Poor (*n* = 475)	Good (*n* = 557)	*p*-Value
		*n (%)*	*n (%)*	
Age	20–29	118 (44.9)	145 (55.1)	<0.00
	30–39	276 (43.8)	354 (56.2)	
	40–49	81 (58.3)	58 (41.7)	
Education	≤High school	116 (50.0)	106 (50.0)	<0.00
	College	138 (47.6)	139 (52.4)	
	≥University	221 (41.0)	312 (59.0)	
Economic Activity	No	202 (39.1)	271 (60.9)	<0.00
	Yes	273 (48.3)	286 (51.7)	
Residence	Rural	165 (50.7)	159 (49.3)	<0.00
	Urban	310 (43.1)	398 (56.9)	
Depression	No	434 (43.3)	576 (56.7)	<0.00
	Yes	41 (59.4)	21 (40.6)	
Fall	No	410 (42.2)	515 (57.8)	<0.00
	Yes	65 (60.8)	43 (39.2)	
Hypertension	No	455 (43.5)	555 (56.5)	<0.00
	Yes	20 (93.1)	2 ( 6.9)	
Diabetes	No	461 (44.1)	550 (55.9)	0.12
	Yes	14 (57.8)	7 (42.2)	
Exercise	No	445 (45.2)	512 (54.8)	<0.00
	Yes	30 (33.4)	45 (66.6)	
Smoking	Non-smoker	426 (43.4)	522 (56.6)	<0.00
	Quit	41 (53.4)	30 (46.6)	
	Yes	8 (74.8)	5 (25.2)	
Drinking	No	265 (44.0)	311 (56.0)	0.59
	Yes	2210 (44.9)	246 (55.1)	
Sleep(h)	0~7	246 (48.1)	253 (51.9)	<0.00
	≥8	229 (40.8)	304 (59.2)	

**Table 3 healthcare-09-01117-t003:** Factors affecting the SRH of pregnant women.

Variables	Categories	OR *	95% CI **
Age	20–29	1	
	30–39	0.86	0.73–1.00
	40–49	0.46	0.35–0.60
Education	≤High school	1	
	College	0.91	0.73–1.15
	≥University	1.13	0.95–1.40
Economic Activity	No	1	
	Yes	1.59	1.37–1.85
Residence	Rural	1	
	Urban	1.35	1.17–1.56
Depression	No	1	
	Yes	0.59	0.44–0.78
Fall experiences	No	1	
	Yes	0.46	0.37–0.56
Hypertension	No	1	
	Yes	0.09	0.02–0.42
Diabetes	No	1	
	Yes	1.08	0.61–1.89
Exercise	No	1	
	Yes	1.92	1.53–2.42
Smoking	Non-smoker	1	
	Quit	0.74	0.54–0.97
	Yes	0.55	0.19–1.61
Drinking	No	1	
	Yes	0.93	0.81–1.08
Sleep(h)	0~7	1	
	≥8	1.25	1.10–1.43

* OR: odds ratio; ** CI: confidence interval.

## Data Availability

Data available in a publicly accessible repository that does not issue DOIs Publicly available datasets were analyzed in this study. This data can be found here: https://chs.kdca.go.kr/chs/rdr/rdrInfoProcessMain.do (accessed on 25 May 2021).

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
