# Peer review of "The Effect of Physical and Mental Health and Health Behavior on the Self-Rated Health of Pregnant Women"

_healthcare, 2021, doi:10.3390/healthcare9091117_

Round 1

Reviewer 1 Report

Relevant work and very pleasant to read. Recommendations on lines of action and future work are equally relevant.

As for the results, there is only doubt about the meaning of the p-value associated with the variable "smoking", which includes pregnant smokers, pregnant women who stopped smoking and pregnant women who did not smoke. Considering that this last group is very small, I don't think there are any conclusions to be observed regarding their health, and I suggest reviewing these same results.

Author Response

We could see the improvement after revising the paper as you advised. Thank you for your valuable comments to improve our paper. Also, it cheered us up when you mentioned what we did. As you suggested, we have revised.
We hope you are always healthy even in the COVID-19 era.

Thank you again!

Reviewer 2 Report

  • In the introduction please restate the sentence implying that childbearing age has recently increased. Historically and worldwide women have been having children throughout the full reproductive period.
  • Please state the differences between college and university
  • Is there a reference for the choice of the economic activity question? It seems a little vague.
  • Was rural or urban, self-reported or based on a local denominator? If it was self-reported was the answer validated by the study team?
  • Perhaps the "poor" group can be renamed to "less than good", especially since the majority of women in that group have fair health. Along this same note, why were women with fair health not included in the "good" group? Fair is usually a lot closer to Good than it is to Poor. Is there a reference for this demarcation?
  • It does not appear as a validated questionnaire was used to assess depression. If this is indeed the case, why did the authors choose this question?
  • How does a fall experience equate to physical health? Where is this validated?
  • Is the 10minutes of exercise for 1 day/week recommended during pregnancy? Where did this value/intensity/frequency come from?

Author Response

(The authors gave the same response as above.)

Reviewer 3 Report

The manuscript "The effect of physical and mental health and health behavior on the self-rated health of pregnant women" is interesting, useful, well designed, executed and written. Congratulations to the authors!

The manuscript by Eunwon Lee and Jiyoung Song is indicated for publication in Healthcare.

Author Response

(The authors gave the same response as above.)
